# Clinical Decision Support System to Detect the Occurrence of Ventilator-Associated Pneumonia in Pediatric Intensive Care

**DOI:** 10.3390/diagnostics13182983

**Published:** 2023-09-18

**Authors:** Jerome Rambaud, Masoumeh Sajedi, Sally Al Omar, Maryline Chomtom, Michael Sauthier, Simon De Montigny, Philippe Jouvet

**Affiliations:** 1Pediatric Intensive Care Unit, Sainte-Justine Hospital, Montreal, QC H3T 1C5, Canada; michael.sauthier.med@ssss.gouv.qc.ca (M.S.); philippe.jouvet.med@ssss.gouv.qc.ca (P.J.); 2Pediatric and Neonatal Intensive Care Unit, Armand-Trousseau Hospital, Sorbonne University, 75012 Paris, France; 3Research Center, Sainte-Justine Hospital, Montreal, QC H3T 1C5, Canada; masoumeh.sajedi.hsj@ssss.gouv.qc.ca (M.S.); sally.alomar@hotmail.com (S.A.O.); simon.de.montigny@umontreal.ca (S.D.M.); 4Pediatric Intensive Care Unit, Caen University Hospital, 14000 Caen, France; chomtom-m@chu-caen.fr; 5School of Public Health, Montréal University, Montreal, QC H2X 3E4, Canada

**Keywords:** pneumonia, ventilator associated, clinical decision system, PICU

## Abstract

Objectives: Ventilator-associated pneumonia (VAP) is a severe care-related disease. The Centers for Disease Control defined the diagnosis criteria; however, the pediatric criteria are mainly subjective and retrospective. Clinical decision support systems have recently been developed in healthcare to help the physician to be more accurate for the early detection of severe pathology. We aimed at developing a predictive model to provide early diagnosis of VAP at the bedside in a pediatric intensive care unit (PICU). Methods: We performed a retrospective single-center study at a tertiary-care pediatric teaching hospital. All patients treated by invasive mechanical ventilation between September 2013 and October 2019 were included. Data were collected in the PICU electronic medical record and high-resolution research database. Development of the clinical decision support was then performed using open-access R software (Version 3.6.1^®^). Measurements and main results: In total, 2077 children were mechanically ventilated. We identified 827 episodes with almost 48 h of mechanical invasive ventilation and 77 patients who suffered from at least one VAP event. We split our database at the patient level in a training set of 461 patients free of VAP and 45 patients with VAP and in a testing set of 199 patients free of VAP and 20 patients with VAP. The Imbalanced Random Forest model was considered as the best fit with an area under the ROC curve from fitting the Imbalanced Random Forest model on the testing set being 0.82 (95% CI: (0.71, 0.93)). An optimal threshold of 0.41 gave a sensitivity of 79.7% and a specificity of 72.7%, with a positive predictive value (PPV) of 9% and a negative predictive value of 99%, and with an accuracy of 79.5% (95% CI: (0.77, 0.82)). Conclusions: Using machine learning, we developed a clinical predictive algorithm based on clinical data stored prospectively in a database. The next step will be to implement the algorithm in PICUs to provide early, automatic detection of ventilator-associated pneumonia.

## 1. Introduction

Ventilator-associated pneumonia (VAP) is a common and severe complication in intensive care units. VAP, as a care-related complication leads to a worsening prognosis for the affected patients and its early diagnosis remain an ongoing challenge in intensive care. In an attempt to enhance VAP detection, the Centers for Disease Control (CDC) issued diagnosis criteria allowing the identification of VAP after 48h of clinical alteration (defined by worsening gas exchange, fever >38 °C or hypothermia, leukocytosis >15,000/mm^3^ or leukopenia <4000/mm^3^, new onset of purulent sputum, apnea or tachypnea, wheezing/rales/rhonchi, cough and bradycardia <100/min or tachycardia >170/min) [1]. However, delays in VAP diagnosis and, to some extent, in initiating anti-infectious therapy are observed and associated with worse outcomes [2,3,4]. Furthermore, subjective criteria included in the CDC pediatric definition for VAP results in a variability of VAP diagnosis and incidence (changes in the appearance and amount of sputum, worsening of an existing cough) [5,6,7]. To help physicians to prospectively diagnose VAP, the CDC developed the concept of Ventilator-Associated Events (VAE) in adults, but children have long been excluded from this definition [1]. It is usual that for adult recommendations, children are excluded mainly because of physiological differences between populations (normal respiratory parameters for an adult are very different from those of a child). Cirulis et al. [8] proposed a pediatric modified VAE definition. Chomton et al. [9] evaluated the pediatric modified VAE definition to detect VAP, but the sensitivity (66%) to identify this ICU-related complication remained disappointing.

In recent years, the number of publications dealing with the development of computerized clinical decision support systems (CDSS) to improve disease diagnosis increased and was shown to be useful for several disease in ICUs [10,11,12,13,14]. The emergence of high-resolution databases supports these developments [15] which allow for a precise and continuous analysis of clinical and biological parameters. Leisman et al. [16] recently reported several recommendations for the development and reporting of predictive models. They identified two categories of predictive models: (1) clinical prediction models for bedside use, and (2) other prediction models intended for deployment across populations for research, benchmarking, and administrative purposes. The usefulness of CDSS had already been highlighted by Mack et al. [17] but no reports on VAP are available currently. To that effect, our project has been developed with the main objective of developing a predictive model to provide early diagnosis of VAP at the bedside in a pediatric intensive care unit (PICU).

## 2. Materials and Methods

This single-center retrospective study was performed using the data collected in the PICU electronic medical record (Intelligence Critical Care and Anesthesia (ICCA^®^); Philips Medical, version F0.1) of a tertiary-care pediatric teaching hospital (Sainte-Justine Hospital, Montréal, QC, Canada). To improve data quality, ICCA^®^ was configured with drop-down menus and critical values alerts. Furthermore, all data entered in ICCA^®^ benefited from a medically-endorsed validation.

The hospital database was queried using SQL Server Management Studio 18^®^ (Microsoft, Redmond, WA, USA) to select patients who were aged from 1 day to 18 years at PICU admission and were mechanically ventilated for more than 48 h, between September 2013 and October 2019. We analyzed the first 30 days of invasive mechanical ventilation.

During the first step of the study, all medical files were reviewed by two senior pediatric intensive care experts (JR and PJ) to classify patients into two groups: VAP patients and free-of-VAP patients. VAP was defined according to the 2021 CDC criteria [1]: The 1st context criteria: invasive mechanical ventilation for more than 48 h, 2nd radiological criteria: new or progressive and persistent infiltrate/consolidation/cavitation, 3rd clinical criteria: worsening gas exchange, fever >38 °C or hypothermia, leukocytosis >15,000/mm^3^ or leukopenia <4000/mm^3^, new onset of purulent sputum, apnea or tachypnea, wheezing/rales/rhonchi, cough and bradycardia <100/min or tachycardia >170/min. 

The second step of the study consisted in the extraction of data coming from the electronic medical record (ICCA^®^, Philips, Toronto, ON, Canada) and high-resolution database (database collecting and storing data from medical devices in real time) [15]. The queried data were date, time, weight (kg), white blood cell count (/mm^3^), neutrophil count (/mm^3^), partial pressure of carbon dioxide (PaCO_2_ in mmHg), partial pressure of oxygen (PaO_2_ in mmHg), inspired fraction of oxygen (FiO_2_ in %), positive end-expiratory pressure (PEEP in cmH_2_O), peak inspiratory pressure (PIP in cmH_2_O), mean airway pressure (MAwP in cmH_2_O), respiratory rate (/rpm), tidal volume (mL), subjective amount of respiratory tract secretion (0, +, ++, +++), oxygenation (OI) and oxygen saturation index (OSI) [18], calculated pulmonary dynamic compliance (in barometric ventilation mode: tidal volume/(PIP–PEEP); and in volumetric ventilation mode: tidal volume/(peak pressure–PEEP)). We also gathered PIM 2 [19] and PELOD-2 scores [20,21].

*Data formatting.* The data was formatted using R (version 3.6.1) as a preparation step to train the prediction models based on different algorithms.

All times were expressed as a relative duration since ICU admission.

*Data cleaning and Missing data.* Incoherent data were identified and corrected according to the scheme described in Appendix A. Variables consisting of data streams of continuous values were imputed following the last observation carried forward method. For missing data at the beginning of the stream, the first valid observation was carried backward.

*Segmenting Variables in Time Blocks.* The variables data streams were first segmented into time blocks of 6 h and then for each variable the median (mode for the discrete variable) was calculated over each 6 h time block to avoid aberrant or missing data. Then, the 6 h blocks were aggregated into 48 h time blocks. We chose to aggregate into 48 h time blocks to be as close as possible to the actual VAP timing definition. For each variable, two columns were generated. One consisted of the first non-missing value among the 6 h time blocks and the other one the last non-missing value among the 6 h time blocks, if there was any, in each 48 h time block (if there was no observation, the data was considered missing). For the development of the algorithms, for each variable, the first non-missing values and the actual difference or relative change of the values of the two columns were considered (more details are available in Appendix A). 

*Stratified train-test split at a patient level.* VAP patients and non-VAP patients were split into the training set (70% of each class) and the testing set (remaining 30% of each class). Since some patients had more than one stay in the PICU, all stays of a patient in the training set were kept in the training set (and the same for patients in the testing set). All details for the train-test split are available in Appendix A. 

*Imputation.* Preliminary inspection of the dataset showed that around 50% of data was missing for the variables “pulmonary dynamic compliance” and “minute ventilation”. Missing values imputation in the training dataset was performed by ‘randomForest’ (v4.6-14) with the function ‘rfImpute’ [22]. The imputed values were the weighted average of the non-missing observations, where the weights were the proximities from randomForest. For data in the testing set, the missing values in each variable were replaced by the mean of the imputed values for the variables with missing values in the training set (more details are available in Appendix A).

*Predictive models.* We applied six different learning algorithms to generate predictive models. The algorithms were: Random Forest with the function ‘rfsrc’ and error rate as the measure of performance, Imbalanced Random Forest with the function ‘imbalanced’ and G-means as the measure of performance, Stepwise Regression and Random Forest using 5-fold cross validation (5-CV) with the ‘train’ function; ‘glmStepAIC’ and ‘rf’ methods and accuracy were used to select the optimal model using the largest value [23]. Finally, we implemented Elastic Net Regression (5-CV) and Weighted Elastic Net Regression (5-CV) with the ‘glmnet’ method and ROC was used to select the optimal model using the largest value. The hyperparameters for the Random Forest, Imbalanced Random Forest and stepwise regression (5-CV) algorithms were ‘ntree’ (number of trees used at the tuning step) and ‘mtry’ (number of variables randomly selected as candidates for the division of a node) [24]. The parameters in Elastic Net regression were alpha, which controls the relative balance between the lasso and ridge regularization, and lambda, which controls the amount of the penalty. All these models used readily available implementations in R [25,26]. Here, cross-validation was performed inside the training set only (more details are available in Appendix A).

*Performance measure and model choice.* Models resulting from the different algorithms were evaluated, at the level of 48 h time blocks, on the train and the test set by calculating their AUC score and by determining classification thresholds reaching predetermined levels of sensitivity (80%, 85%, 90%, 95%). The final model was chosen based on the capacity to [1] maximize specificity under these sensitivity levels, and [2] generalize the sensitivity and specificity from the test set. The area under the ROC curve (AUC) was considered as the primary measure of performance to choose the best model.

*Per patient validation.* The final model was evaluated on its capacity to correctly assess the infection status of patients over time. The predictions’ results obtained after setting different classification thresholds were taken. The number of patients with accurate predictions (i.e., predicted class = observed VAP status) and inaccurate predictions (i.e., predicted class ≠ observed VAP status) were computed over time. The number of patients for whom the predictions contained at least one error were identified. We looked at the accuracy of predictions by stratifying patients into two groups. We identified the patients for whom the predictions contained at least one error for each subgroup. The global error rates were calculated for each subgroup. 

### Statistics

Development of the clinical decision support was performed using open-access R software, Version 3.6.1^®^ (R Foundation for Statistical Computing, Vienna, Austria). Statistical analysis of patients’ characteristics was performed using Prism X^®^ software (version 7.05) (GraphPad Inc. San Diego, CA, USA). Kolmogorov analysis was performed to test the normal distribution of continuous variables. Population description used categorical variables expressed as frequency with corresponding proportion and quantitative variables presented as mean and standard deviation. Performance evaluation was conducted using ROC curves, AUC and their confidence intervals, and derived measures of sensitivity and specificity. The ethical committee of Sainte-Justine University hospital approved the study and waived the need for informed consent given the retrospective design.

The Saint-Justine ethical committee approved the study as a retrospective study and waived the need for written consent (n°2020–2454).

## 3. Results

### 3.1. General Description of the Population

A total of 5153 children had been hospitalized in Saint-Justine PICU during the study period of which 40% (2077) were mechanically ventilated and 1235 episodes with more than 48 h of mechanical invasive ventilation were identified (Figure 1). Seventy-seven patients had at least one VAP event. Seventy-eight VAP events (6%) were diagnosed by two experts. The patients’ general characteristics are described in Table 1.

Patients with less than 4 days of mechanical ventilation were removed (see Appendix A) to achieve 811 episodes of invasive mechanical ventilation. The training set (70% of each class) and testing set (remaining 30% of each class), resulted in a training set of 461 patients free of VAP and 45 patients with VAP and in a testing set of 199 patients free of VAP and 20 patients with VAP. Since some patients had more than one stay in the ICU, there could be different events for the same patient. The training set thus had 513 stays with no VAP event and 45 stays with a VAP event, and the testing set had 231 stays with no VAP event and 22 stays with a VAP event. The segmenting of variables in 48 h non-overlapping time blocks generated, from these datasets, 1852 time blocks free of VAP and 45 time blocks with VAP in the training set, and 788 time blocks free of VAP and 22 time blocks with VAP in the testing set.

We observed similar characteristics in the train and test groups (Table 2).

### 3.2. Missing Data

We observed two missing values for “sf ratio” and “oxygen saturation index (OSI)” in the test set (0.1% of total observations). For the variable “pulmonary dynamic compliance” the proportion of missing values in the train and test sets were 0.49 and 0.54, respectively. For the variable “minute ventilation”, the proportion of missing values in the train and test sets were 0.49 and 0.54, respectively.

### 3.3. Results of Training Algorithm

The Imbalanced Random Forest model was considered as the best fit with an area under the ROC curve of 0.86 from the train set.

Thresholds and specificities corresponding to the predetermined levels of sensitivity are presented in Table 3. Variable importance obtained from the Imbalanced Random Forest model are presented in Figure 2.

### 3.4. Performance on Test Dataset

The area under the ROC curve from fitting the Imbalanced Random Forest model on the test set was 0.82 (95% CI: (0.71, 0.93)) (Figure 3).

The specificity and sensitivity obtained after setting different classification thresholds are presented in Table 4. An optimal threshold of 0.41 gave a sensitivity of 79.7% and a specificity of 72.7%, with a positive predictive value (PPV) of 9% and a negative predictive value of 99%, with an accuracy of 79.5% (95% CI: (0.77, 0.82)).

### 3.5. Per Patient Validation

Performance of the final model was evaluated over different time periods. Time periods were defined starting from the first time block and going up to a given time block in the future. The confusion matrices for all the time periods were constructed. False positive rates (FPR), true positive rates (TPR), and area under the curve (AUC) were calculated. The results are presented in Figure 4. The procedure is explained in detail in Appendix A.

The global error rate is presented in Table 5. We observed a lower error rate for patients with at most three time blocks of observations, compared to the ones with at least four time blocks of observations.

## 4. Discussion

Using an electronic medical record, an algorithm supporting clinicians in the early diagnosis of ventilator-associated pneumonia in PICU had a sensitivity of 80% and specificity of 73%, with the threshold of 0.41. To date, it is the most accurate sensitivity achieved by a CDSS system to provide early detection of VAP.

Ventilator-associated pneumonias is a severe health care disease [2,27,28]. To improve the delay and accuracy of this challenging diagnosis, Cirulis et al. [8] evaluated the accuracy of adults’ ventilator-associated events (VAE) to early diagnose pediatric VAP and developed modified pediatric criteria for VAE (increase in FiO_2_ by 20% or PEEP by 2 cm H_2_O sustained for more than one day). VAE and modified pediatric VAE both had a disappointing sensitivity of 23% and 56% for Cirulis et al. [8] and 56% and 66% for Chomton et al. [9], respectively. Our algorithm was based on machine learning methods and improved the sensitivity in this study and could be implemented to screen in real-time patient’s data to provide early detection of VAP in children. The prediction of the test set using the Imbalanced Random Forest model is stored in a file and is available on Github [29].

Implementation of a clinical decision system to help physicians is a promising technology aimed at helping the physician to take medical decision [10,30], to analyze chest X-rays [31], or to increases diagnosis sensitivity [32]. The development methodology starts with a retrospective classification of analyzed patients to define whether they develop the studied conditions (e.g.,VAP). This step is crucial to develop an accurate algorithm and rely on the quality of the classification method. In a large review of published CDSS, Ostropolets et al. [33] highlight that only one manuscript addressed confounding and bias due to misclassification. Our classification methodology included all the relevant data from the electronic medical record clinically collected and is the best accuracy that can be obtained currently.

In addition to the classification methodology, the main strength of this study includes the use of continuous vital signs and the ventilatory parameters monitoring database, limiting the number of missing data and allowing the use of the algorithm in real time in the future [15]. The variables extracted from this monitoring included the OSI ratio, the variation of pulmonary compliance, minute ventilation, and ventilatory median pressures. However, the algorithm identified the variation of PEEP during the last 48 h preceding the VAP as the most important criteria as suggested by the CDC definition. Nevertheless, the variation of the ventilatory mean airway pressure was the second most important variable. This result seems crucial because the ventilatory mean airway pressure that not only includes PEEP but also the PIP, I/E ratio and instantaneous gas flow is not included in the CDC diagnosis criteria for VAP.

Nonetheless, we noticed that our algorithm has a better efficacy to diagnose early VAP (before day 6 of the PICU stay) versus late VAP (after day 6 of the PICU stay), with the error rate in prediction of 23.08% vs. 66.67%, respectively. We can hypothesize that the more time the patient stays in the PICU, the more discrete are the variations to be detected due to the potential alteration of the patient’s condition.

This study has several strengths. First, this is the first study with a CDSS system reaching over 80% sensitivity. Second, despite this being a single-center study, we report one of the largest number of patients included in a study in children. Finally, we report the highest sensitivity and specificity to diagnose VAP.

Despite these promising results, this work suffers from several limitations. First, the invasive procedures were not considered in our algorithm (bronchoscopy, transportations) due to the lack of data concerning the timing between these procedures and the VAP. Second, data on the reason for invasive mechanical ventilation were not reported in all medical files, although it is well known that brain injury and neurological disorder with impaired swallowing predispose more to pneumonia. Third, the treatment of missing data was conducted using data-focused approaches (last observation carried forward for missing data mid-stream, first observation carried backward for data missing at the beginning of a stream) which did not model the missing data process; the classifications between the VAP and non-VAP patients were retrospectively performed which may have resulted in some misinterpretation of the clinical data. Fourth, for generalizability, a prospective validation of the algorithm in several PICU needs to be conducted. 

## 5. Conclusions

We developed the first clinical predictive system dedicated to VAP diagnosis in PICUs using a high-fidelity database. The implementation of such an algorithm in PICUs could allow physicians to be alerted early in cases of respiratory function impairment and to decide whether to perform respiratory tract analysis and start anti-infective treatment. Although this algorithm achieves a promising sensitivity and specificity level, it is still lacking power and cannot be implanted in PICUs. Additionally, it still needs to be prospectively validated in other PICUs to confirm its reproductivity and external power.

## Figures and Tables

**Figure 1 diagnostics-13-02983-f001:**
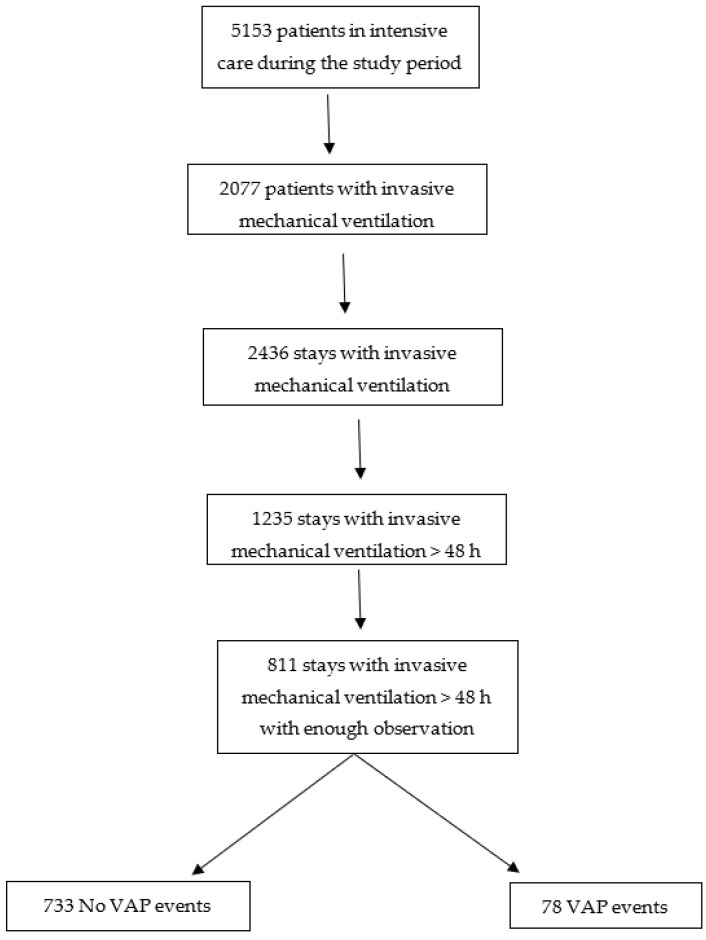
Flow chart. VAP: Ventilator-associated event.

**Figure 2 diagnostics-13-02983-f002:**
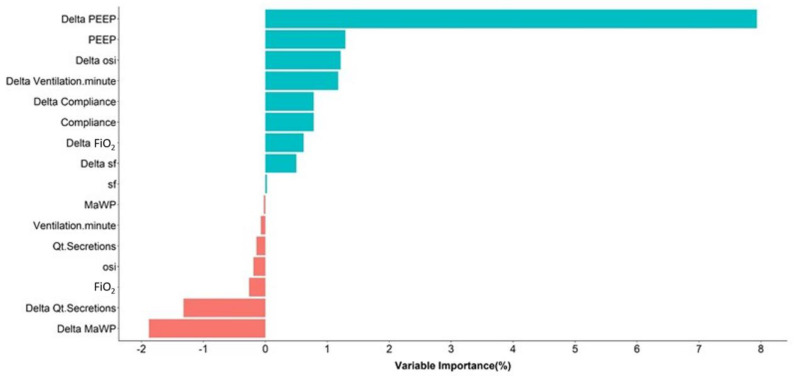
Variable importance used in the clinical decision system.

**Figure 3 diagnostics-13-02983-f003:**
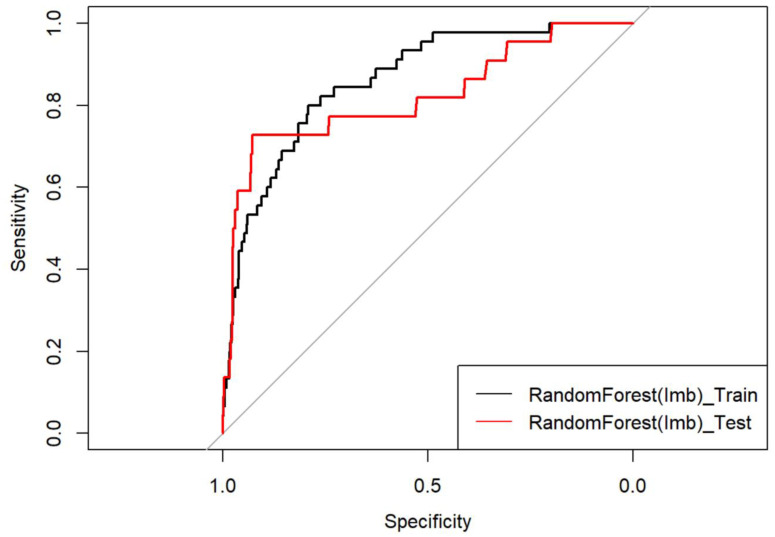
ROC Curve. Black curve represent the efficiency of the training of the algorithm on 2/3 of the dataset. Red curve represents the efficiency of the test of the algorithm on the rest of the data set.

**Figure 4 diagnostics-13-02983-f004:**
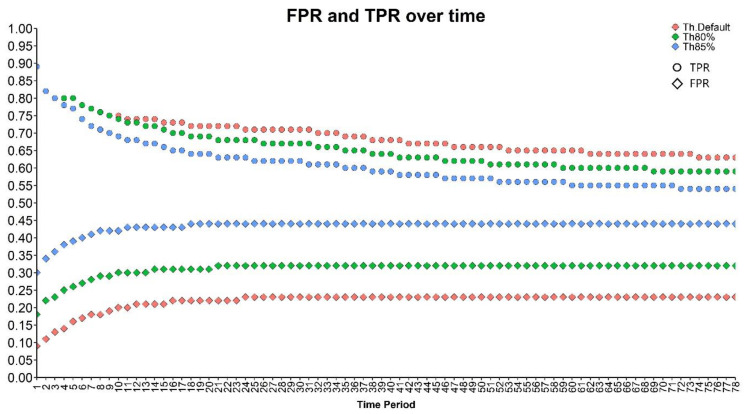
False positive rate and true positive rate over different time periods for different thresholds. Th.Default: default threshold of the model; Th80%: threshold correspond to the 80% sensitivity; Th85%: threshold correspond to the 85% sensitivity.

**Table 1 diagnostics-13-02983-t001:** Population characteristics.

Population Characteristics	Global Population (*N*: 827)	VAP Patients (*N*: 77)	No VAP Patients (*N*: 750)	*p*:
Weight (kg)	15.8 ± 1.6	20.99 ± 2.7	15.25 ± 0.7	0.01
Age (days)	1308 ± 1904	1806 ± 250	1256 ± 69	0.02
Gender male (%)	475 (57%)	41 (53%)	434 (58%)	0.4
Pelod 2 score	10.1 ± 4.8	10.4 ± 0.6	9.9 ± 0.2	0.47
Pelod 2 mortality risk (%)	0.3 ± 0.3	0.3 ± 0.1	0.2 ± 0.01	0.15
Bronchoscopie (%)	70 (8%)	14 (18%)	56 (8%)	0.04
Neuromuscular blocker (%)	279 (34%)	43 (55%)	236 (31%)	<0.0001
Mechanical Ventilation duration (days)	12.5 ± 30.9	29.3 ± 5.1	10.9 ± 1.5	<0.0001
PICU length of stay (days)	26.1 ± 52.5	48.3 ± 7.1	23.4 ± 1.8	<0.0001
Survival rate (%)	740 (90%)	65 (84%)	675 (90%)	0.16

PICU: Pediatric intensive care unit; VAP: Ventilator-associated pneumonia.

**Table 2 diagnostics-13-02983-t002:** Train and test groups’ characteristics.

Train and Test Groups Characteristics	Test Group (*n*: 261)	Train Group (*n*: 572)	*p*:
Weight (kg)	16.9 ± 1.3	15.6 ± 0.8	0.40
Age (days)	1387 ± 129	1268 ± 84	0.43
Gender male, (*n*, %)	146 (60)	284 (58)	0.69
Pelod 2 score	10.4 ± 0.2	9.7 ± 0.5	0.16
Pelod 2 mortality risk (%)	0.3 ± 0.1	0.2 ± 0.1	0.13
Proportion of VAP patients (*n*, %)	25 (10)	50 (10)	0.99
Length of mechanical ventilation before VAP (days)	9.9 ± 2.7	9.6 ± 1.9	0.66
Length of mechanical ventilation duration (days)	12.1 ± 1.6	11.2 ± 1.1	0.64
PICU length of stay (days)	21.3 ± 2.4	22.3 ± 2.0	0.81

Pelod: Pediatric logistic organ dysfunction, PICU: Pediatric intensive care unit; VAP: Ventilator-associated pneumonia.

**Table 3 diagnostics-13-02983-t003:** Imbalanced Random Forest model. Threshold and specificity from predetermined sensitivity for the train set.

Threshold	Specificity	Sensitivity
0.41	0.79	0.80
0.29	0.64	0.87
0.25	0.58	0.91
0.22	0.52	0.96

**Table 4 diagnostics-13-02983-t004:** Imbalanced Random Forest model. Sensitivity and specificity for the test set corresponding to different thresholds.

Threshold	Specificity	Sensitivity
0.41	0.797	0.73
0.28	0.66	0.77
0.25	0.59	0.77
0.22	0.53	0.82

**Table 5 diagnostics-13-02983-t005:** Error rates (%) for predicted classes.

		G1			G2	
	ER.Pred	ER.Pred.th80	ER.Pred.th85	ER.Pred	ER.Pred.th80	ER.Pred.th85
All	11.56	19.60	31.66	79.59	83.67	95.92
VAP	23.08	23.08	23.08	66.67	66.67	88.89
NoVAP	10.75	19.35	32.26	82.50	87.50	97.50

G1: Patients with at most 3 time blocks of observations; G2: Patients with at least 4 time-blocks of observations. E.Pred: Error rate for prediction; E.Pred.the80%: Error rate for prediction with threshold correspond to the 80% sensitivity; E.Pred.th85%: Error rate for prediction with threshold correspond to the 85% sensitivity.

## Data Availability

Data are available on demand.

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
