# Peer review of "Clinical Decision Support System to Detect the Occurrence of Ventilator-Associated Pneumonia in Pediatric Intensive Care"

_diagnostics, 2023, doi:10.3390/diagnostics13182983_

Round 1

Reviewer 1 Report

The authors, J. Rambaud et al, describe in their work, through the collection of data from several years on intubation pneumonia in hospitalised children, an important prognostic diagram and method for resolving and nullifying such events.

It is a well-articulated work with only a few minor editing errors.

It should only better detail the limitations due to the absence of the medical history that may influence the choice of assisted ventilatory need for more than 48 hours.

In other words, some medical conditions, such as neurological adverse events or trauma may impose more time on assisted ventilation and thus predispose more to pneumonia. But other than that I think it is an excellent work and I congratulate the authors.

Reviewer 2 Report

The current study titled “Clinical decision support system to detect the occurrence of 2 ventilator associated pneumonia in pediatric intensive care” Ref: diagnostics-2601518, deals with an important subject. It is well explained. Minor revisions are needed considering the following items.

- A list of abbreviations used throughout the entire study should be added.

- The conclusion section should be revised explaining the fruitful observations attained by the study.

Reviewer 3 Report

My concern about this manuscript is given below-

Title and Abstract is okay 

The introduction section effectively establishes the contextual background about the necessity for enhancing diagnostic criteria for ventilator-associated pneumonia (VAP), particularly within pediatric healthcare settings.

Introduction clarification and language suggestion is given below

ine 43-44: It could be helpful to briefly describe the nature of the “clinical alteration” that triggers the CDC diagnosis criteria after 48h.

Lines 46-47: Given the mention of "subjective criteria," a brief elucidation of one or two such criteria might provide more context to readers unfamiliar with the topic.

Lines 48-51: The introduction of VAE and its exclusion from pediatric criteria is an essential point, but it may benefit from a concise explanation of why children were initially excluded.

and 

Line 40: Consider rephrasing "This care-related complication" to "VAP, as a care-related complication, ..."

Line 42: "To improve the detection of VAP," could be more fluid as "In an attempt to enhance VAP detection,"

Line 56-57: The phrase "These developments are supported by the emergence of high-resolution databases" might read smoother if restructured: "The emergence of håigh-resolution databases supports these developments

Material methods: Okay but still need more details of Data cleaning and Missing data

Figure need more detailed legends 

Conclusion: Author should rewrite and elaborate in term of following sentances -This algorithm achieves promising sensitivity and specific level but needs to be 367 prospectively validated (programed next step).

Minor revision required
